# Personalized Diet With or Without Physical Exercise Improves Nutritional Status, Muscle Strength, Physical Performance, and Quality of Life in Malnourished Older Adults: A Prospective Randomized Controlled Study

**DOI:** 10.3390/nu17040675

**Published:** 2025-02-13

**Authors:** Huzeyfe Arıcı, Yavuz Burak Tor, Mustafa Altınkaynak, Nilgün Erten, Bulent Saka, Osman F. Bayramlar, Zeynep Nur Karakuş, Timur Selçuk Akpınar

**Affiliations:** 1Department of Internal Medicine, Division of Geriatric Medicine, Cerrahpaşa Faculty of Medicine, Istanbul University, 34098 Istanbul, Turkey; 2Department of Internal Medicine, Memorial Bahcelievler Hospital, 34180 Istanbul, Turkey; yavuzburaktor@gmail.com; 3Department of Internal Medicine, Faculty of Medicine, Istanbul University, 34093 Istanbul, Turkey; mustafa.altinkaynak@istanbul.edu.tr (M.A.); snilgunerten@hotmail.com (N.E.); drsakab@yahoo.com (B.S.); doktortimur@gmail.com (T.S.A.); 4Occupational Physician, Enerjisa Energy AS, 34746 Istanbul, Turkey; obayramlar@gmail.com; 5Food Technology Program, Eşme Vocational School, Uşak University, 64000 Uşak, Turkey; zeynep.karakus@usak.edu.tr

**Keywords:** malnutrition, aged, diet, exercise, quality of life

## Abstract

**Objectives:** Malnutrition (MN) is prevalent in older adults and closely related to sarcopenia, frailty, morbidity, mortality, and decreased quality of life. In this study, we aimed to evaluate the effects of a personalized diet combined with planned physical exercise on nutritional status, physical performance, and quality of life (QoL) in malnourished older adult patients. **Methods:** In this prospective study, 20 older adults with MN risk according to the Mini Nutritional Assessment—Short Form (MNA-SF) were randomized into (i) personalized diet (PD) and (ii) personalized diet with physical exercise (PDE) groups, and followed up with for 12 weeks. The physical exercise included warm-up, strengthening, balance, and cooldown phases, with a frequency of 3–4 days per week. Anthropometric measurements, physical performance, and quality of life were assessed using standardized tools at baseline and at the 4th, 8th, and 12th weeks. QoL was measured using the EQ–5D index and EQ–5D visual analog scale (VAS) scores. **Results:** A total of 20 patients (55% male) participated in the study. During the study, BMIs, MNA-SF scores, and hand grip strength were increased, and the patients’ average duration on the Timed Up and Go (TUG) test decreased significantly in both groups. The EQ–5D index score of the PD group and the EQ–5D VAS scores of both groups were increased. **Conclusions:** A personalized diet with or without physical exercise therapy was associated with improved nutritional status, physical performance, and QoL.

## 1. Introduction

Malnutrition (MN) and sarcopenia are the main reasons for frailty in older adults, which is closely associated with immobility, falls, dependency, infections, hospitalization, and mortality [1,2,3]. Current evidence indicates that adequate energy and protein intake, physical exercise (particularly resistance training), and vitamin D supplementation should be the main steps in the treatment of MN and sarcopenia [4,5,6]. Marcos-Delgado et al. indicated that adherence to high-quality diets (such as the Mediterranean diet and the Alternative Healthy Eating Index-2010) was associated with a lower prevalence of MN in older adults. Specific components such as a higher consumption of fish, long-chain n-3 fatty acids, vegetables, and legumes can reduce MN risk. Given the differences in lifestyle, disease history, age, and physical activity among individuals, personalized diets are crucial in the treatment of MN. They also noted that low energy and protein intakes increase the risk of MN, particularly in older adults, emphasizing the need for tailored dietary strategies [7]. In another study, adherence to the Mediterranean diet was associated with a reduced risk of MN, which was scored using [8].

MN is not merely a medical condition but a complex issue that intertwines with various dimensions of an individual’s life, significantly affecting their overall quality of life (QoL). MN has a profound impact on quality of life through its effects on physical health, mental well-being, social interactions, and functional abilities. Addressing MN is essential for improving the overall quality of life for individuals affected by it.

Evidence from cohort studies indicated that older individuals with a high risk of MN are more likely to experience a poor quality of life [9]. Another study showed that MN significantly decreased QoL in older adults with rheumatoid arthritis (RA) [10]. Hyunh NTG et al. reported higher QoL in older adults with normal nutritional status when compared to those with MN [11]. A systematic review and meta-analysis including 10 studies demonstrated a significant association between nutritional status and QoL [12].

In this case–control study, we aimed to evaluate the effects of a personalized diet combined with planned physical exercise on nutritional status, physical performance, and quality of life (QoL) in malnourished old aged patients.

## 2. Materials and Methods

### 2.1. Study Population

The study participants were selected from patients admitted to Istanbul University, Istanbul Faculty of Medicine Hospital, Clinical Nutrition and General Internal Medicine Outpatient Clinics. Malnourished old–aged patients (≥65 years old) who were oriented, cooperated, and were independent in daily living activities were included in the study. The exclusion criteria were <65 years of age, cognitive impairment, immobilization, eating/swallowing disorders, receiving enteral or parenteral nutritional support, and having acute medical problems. Patients eligible for the study (*n* = 20) were randomized to the study and control groups according to their dates of admission. While not the most robust form of randomization, using admission dates helped to avoid selection bias by ensuring that all eligible patients were considered for inclusion as they presented, rather than being selectively chosen. Study was retrospectively registered, with clinical trial number NCT06823739 on 12 February 2025.

When evaluating the statistical power of the study, an a priori power analysis was conducted using G*Power (version 3.1). The analysis was based on a repeated-measures ANOVA with a within–between interaction design to determine the required sample size for adequate power. For our observed variable, the power for the within-subject factor (time) was 0.81, with a partial eta squared (η^2^) of 0.07, indicating a medium effect size. For the interaction effect (time × group), the partial eta squared (η^2^) was 0.072, also reflecting a medium effect size. The input parameters included an effect size f = 0.278, alpha error probability (α) = 0.05, desired power (1 − β) = 0.8, two groups (intervention and control), four repeated measurements, and a correlation among repeated measures of 0.5. The nonsphericity correction factor (ϵ) was set to 1. The analysis determined a total required sample size of 20 participants (10 per group), with an actual power of approximately 0.823. These findings suggest that a minimum of 10 participants per group is sufficient to conduct an adequately powered trial (1 − β = 0.8, α = 0.05) in this rehabilitation setting.

A personalized diet combined with a standard planned physical exercise regimen was given to the study group (PDE); the patients in the control group (PD) received a personalized diet only.

### 2.2. Nutritional Status

All the patients were evaluated during the admission (initial visit) and in the 4th, 8th, and 12th weeks. The Mini Nutritional Assessment—Short Form (MNA-SF) test was performed during each visit to evaluate nutritional status. Those who scored 7 points or less on this test were considered malnourished, and those who scored 8–11 points were considered at risk of MN. Patients with MN or at risk of MN were included in the study.

### 2.3. Anthropometry, Bioelectrical Impedance Analysis (BIA), and Muscle Strength Measurements

Body mass index (BMI) was calculated by dividing body weight (kg) by the square of body height (m) and is expressed in kg/m^2^. The appendicular skeletal muscle mass (ASM) was measured through BIA using a Tanita MC 780 MA (Tokyo, Japan). The ASM index (ASMI) was calculated by dividing the ASM by the square of the body height (kg/m^2^). Measurements were performed using the same device in the Clinical Nutrition and Microbiota Research Laboratory, Istanbul Faculty of Medicine. The device was always in the same place during all the visits. During the measurement, it was ensured that the patients had taken off their shoes and socks, were not wearing heavy or tight clothes, did not have moisturizers such as creams on their hands and feet, and were not carrying transmitting devices such as mobile phones. Muscle strength was measured using a standardized handheld dynamometer (Jamar, Duluth, MN, USA), taking the best of 3 measurements made for the dominant hand. In patients who had only one upper extremity or who could use only one extremity, measurements were made with this extremity.

ASMI and muscle strength cut-off values were used (ASMI: <5.86 kg/m^2^ for men and <4.36 kg/m^2^ for women; handgrip strength (HGS): 35 kg for men and 20 kg for women) to define low muscle mass and low muscle strength in our patients [13,14]. Probable and confirmed sarcopenia were diagnosed according to EUGMS2 consensus report definitions [4]. Measurements were taken for each patient at the initial and 4th-, 8th-, and 12th-week visits.

### 2.4. Timed Up and Go Test (TUG)

A TUG test was performed to evaluate physical performance. The patients sat on a standard chair and were then told to get up from the chair, walk to a line 3 m away, come back, and sit on the chair again. Bischoff et al. associated a TUG test result ≥ 20 s with low physical performance [15]. Measurements were recorded at the initial and 4th-, 8th-, and 12th-week visits.

### 2.5. Personalized Diet

The personalized diet was prepared by the two appointed dietitians of the Clinical Nutrition Team of the Hospital, tailored to the patients’ needs (according to daily energy and protein requirements, which were calculated at 25–30 kcal/kg and 1.0–1.2 g/kg, respectively) [16], lifestyle, nutritional preferences, and medical diagnoses. Physiological stress and the physical activity level of the patients were also considered.

In total, 45–55% of the total calories came from carbohydrates, 20–35% from fat, and the diet included 20–30 gr of fiber/day [17].

While preparing the diet plan, the patients’ socioeconomic statuses, palates, and social and psychological factors were considered. The protein sources were diversified, with animal-based proteins (chicken, fish, red meat, and turkey) and plant-based proteins (legumes). They were asked which protein source they usually consumed and which one they preferred in terms of taste. It was explained to patients who had difficulty consuming meat that they could choose legumes, eggs, or dairy products instead of meat. The patients were told to consume fish at least twice a week.

Each breakfast featured eggs and fermented dairy products. The fat content predominantly comprised mono- and polyunsaturated fatty acids, from olive oil, sunflower oil, fish, and nuts. Carbohydrates were selected to have a low glycemic index, favoring whole grains while excluding white flour entirely.

While preparing the diet plan, the people whom the patient lived with were also consulted. It was questioned whether the patient themself or the people they lived with mainly cooked the meals at home. Thus, main meals were discussed with the primary person cooking at home.

The inclusion of fruits and vegetables aimed to maximize the antioxidant diversity and color variety. To enhance calorie intake and dietary adherence, milk-based and fruit-based desserts were incorporated twice weekly. Hydration goals were set at a minimum of 30 mL/kg/day with seasonal variations. For patients struggling to meet these goals, additional fluids were provided through freshly squeezed juices and unsweetened homemade compotes. Tea and coffee consumption was limited to one cup daily. For individuals requiring caffeine restrictions due to conditions such as hypertension, cardiac disease, arrhythmia, or anxiety, decaffeinated coffee was recommended.

Daily salt intake was restricted to 5–6 g, and for patients with previously high salt consumption, herbs and spices were used to enhance flavor and improve adherence. To support meal planning and encourage dietary compliance, participants received education on the “healthy plate model”, promoting variety and balanced nutrition [16]. Diet compliance was monitored through regular phone calls every week and during the visits.

### 2.6. Physical Exercise

The PDE group participants followed a home-based exercise program for 12 weeks, with a frequency of 3–4 days per week, which was planned by the two appointed physiotherapists of the Hospital. The exercise began with a 10-min warm-up session, including posture, stretching, and stationary marching. The intensity of the subsequent exercises was set to a very light level (9–10) based on the Borg Scale, corresponding to a perceived exertion that increased the heart rate to around 40–50% of the maximum. The Borg Scale is a tool used to measure an individual’s perceived level of effort during physical activity. It is based on how hard a person feels their body is working. The scale ranges from 6 to 20, with each number corresponding to a level of intensity. On this scale, 6 means no exertion at all and 20 means maximal exertion [18].

This exercise program in the PDE group was designed to be performed at home without requiring any specialized equipment. It aims to improve physical health, balance, and coordination while maintaining muscle strength and flexibility. Patients performed brisk walking at least three times a week for a duration of 10 to 30 min per session. It also included balance practice and coordination exercises for 10 min. These include standing on tiptoes, balancing on one foot while holding onto a chair for support, and walking in a straight line as if on a narrow beam. Finally, the exercise session ended with a 10-min cooldown, focusing on relaxing muscles and performing gentle stretches to improve flexibility and reduce tension. All of the exercises were both explained and provided in a figured chart, to the patients or the relatives.

The patients’ physical exercise compliance and adherence were monitored through regular phone calls.

### 2.7. QoL 

The EuroQoL—5 Dimensions (EQ–5D) quality-of-life score and visual analog scale (EQ–5D VAS) were used to assess quality of life. EQ–5D is a five-question test for measuring quality of life introduced by the EuroQoL group, the Western European Quality of Life Research Society. Its five dimensions are mobility, self-care, usual activities, pain/discomfort, and anxiety/depression. Each dimension has three levels: no problems, moderate problems, and serious problems. There is also a vertical visual analog scale from 0 to 100 on which patients rate their health (VAS). This scale’s endpoints are labeled “your best health situation” and “your worst health situation”. Thus, 243 possible different health outcomes can be identified with this scale. Based on the five dimensions of the scale, the index score ranged from −0.59 to 1. In this calculation, −0.59 indicates a quality of life worse than death and 1 indicates a perfect quality of life [19].

### 2.8. Statistical Analysis

Quantitative variables are expressed as means, medians, and interquartile ranges (IQRs) for continuous data, and as percentages (%) and frequencies (n) for categorical data. Because the data did not follow a normal distribution, non-parametric tests were used for comparisons.

The Kruskal–Wallis test was applied to compare more than two non-parametric groups. For pairwise comparisons, the Mann–Whitney U test was used as a post hoc test. Spearman’s correlation test was used to measure the correlation between variables, while the chi-square test was used to compare categorical variables. Fisher’s exact test was chosen because it provides accurate results for small sample sizes (*n* < 30) and is particularly suitable when the expected frequencies in contingency tables are less than 5.

All the results were evaluated within a 95% confidence interval, and *p* < 0.05 was considered statistically significant. Statistical analyses were performed using IBM SPSS-21 (Statistical Package for Social Sciences version 21.0).

## 3. Results

A total of 20 malnourished patients aged 65 and above were included in the study (mean age: 71.8 ± 5.4 years; 11 males). The characteristics of the patients are provided in Table 1. The PD and PDE groups did not show any significant differences in terms of gender, age, chronic diseases, or nutritional status (Table 1). The initial MNA-SF scores showed 60% MN and 40% MN risks in both groups (Table 1). After 12 weeks of intervention, the mean BMI of the patients increased from 25.9 kg/m^2^ to 27.6 kg/m^2^ in the PD group, and 29.2 kg/m^2^ to 30.1 kg/m^2^ in the PDE group (Table 2). According to the MNA-SF test results, 70% of the patients in both groups showed improvements in their nutritional statuses in the 4th week, which were 100% in the 12th week, and all of the patients from both groups showed MNA-SF scores ≥ 12 at the end of the study. The median MNA–SF scores of the patients in the PD and PDE groups were increased from 7.1 to 13, and 7.7 to 13.5, respectively (Table 2).

The HGS measurements of the patients were low in 90% of the patients in both groups. They were significantly increased at the end of the study (mean values: 20.9 kg to 27.5 kg in the PD group, and 20.3 kg to 27.3 kg in the PDE group). The mean ASMI was increased from 7.7 kg/m^2^ to 8.4 kg/m^2^ in the PD group, and 8.5 kg/m^2^ to 8.6 kg/m^2^ in the PDE group, but these differences did not reach statistical significance. According to the EWGSOP2 cut-offs, only two patients in the PD group had confirmed sarcopenia during admission, while their 12^th^-week measurements showed improvement with a personalized diet (Table 2).

The mean TUG test results of the patients in the PD and PDE groups were 13.6 ± 2.9 and 15.0 ± 4.9 s, respectively. After 12 weeks of intervention, they were significantly improved to 10.8 ± 1.7 in the PD group and 11.9 ± 2.7 s in the PDE group (Table 2).

The patients’ initial mean EQ–5D index scores were 0.75 in the PD group and 0.7 in the PDE group, and were 0.82 and 0.74 at the end of the study. Although the 9.4% increase in the mean EQ–5D index score in the PD group was statistically significant, the 5.3% increase in the PDE group did not reach statistical significance (Table 2). The mean EQ–5D VAS scores of the patients in the PD and PDE groups were significantly improved in both groups at the end of the study (PD group: 67 to 79; PDE group: 65 to 78) (Table 2).

The EQ5D VAS score and ASM measures of the whole study group were found to be correlated at the end of the study (*p* < 0.001) (Table 3). Moreover, a negative correlation between the EQ5DVAS score and TUG test time changes was found after 12 weeks of intervention (ΔEQ5DVAS vs. ΔTUG test time: *p* = 0.012; correlation coefficient = −0.550).

## 4. Discussion

MN is prevalent among old-aged people [20]. We used the MNA-SF to evaluate the nutritional statuses of our patients. The MNA is used to identify individuals with MN and those at risk of MN among older adults [21]. Accordingly, ESPEN recommends using the full or short form of the MNA to screen nutritional status in older adults [22].

Ordovas and Berciano showed that personalized dietary interventions, informed by individual genetic, epigenetic, and metabolomic profiles, can promote healthy aging. Nutrition advice can be personalized based on individual genetic and biological traits. These strategies can help to address age-related conditions like inflammation, diabetes, and cardiovascular disease. For instance, the Mediterranean diet has been shown to interact beneficially with specific genetic variants. Variations in dietary responses among individuals, influenced by genetic factors, underscore the need for personalized approaches. Personalized nutrition has the potential to mitigate age-related comorbidities and improve quality of life by integrating genomic, epigenomic, and metabolomic data with dietary and environmental factors [23]. Burton et al. discussed strategies for personalized nutrition to meet the specific needs of older adults, emphasizing the importance of personalized approaches to nutrition for improving health outcomes and quality of life. Nutrition plays a crucial role in metabolic health and can improve the quality of life for older adults. Studies aim to develop tools and services to assess individual nutritional needs, creating functional foods personalized to older adults, and utilizing technology for health monitoring and dietary adherence. Personalized nutrition can meet the diverse health and nutritional needs of older adults, resulting in better health outcomes and quality of life [24].

Magzal et al. showed the effects of a personalized diet, based on the Mediterranean diet’s principles, on quality of life, nutritional intake, and changes in gut microbiota among older adults living independently. A 6-month personalized dietary program guided by a registered dietitian was implemented in older adults, and at the end of the study, quality-of-life scores had improved, particularly for psychological well-being. The study demonstrated that a personalized Mediterranean-style diet can improve quality of life and positively alter the gut microbiota in older adults [25].

We observed a statistically significant improvement in the mean MNA-SF scores of the both groups with a personalized diet plan (MNA-SF the scores of our patients were ≥12 at the end of the study).

According to ESPEN guidelines, the normal BMI range for the geriatric population is 20–24.9 kg/m^2^ [26]. A recent meta-analysis indicated that the BMI range of 23–28 kg/m^2^ is most strongly associated with lower mortality and disability, and, as such, may be considered optimal for the geriatric population [27]. Although BMI alone does not indicate overall health status, it plays a significant role in the evaluation of nutritional status. Additionally, it is often increased by an effective nutritional treatment. A significant increase in the mean BMI of our patients indicated that a personalized diet with well-planned monitoring can be used to treat MN in old-aged patients.

In our study, the initial mean BMI and ASMI of the PDE group were higher than those of the PD group, but not in a statistically significant manner (Table 1). Despite the small number of patients, the study provides important insights into the role of diet and exercise interventions in improving health outcomes among older adults. With a higher number of participants, PDE could probably make a detectable and significant difference in terms of muscle mass. However, the results of this preliminary study highlight the importance of a personalized diet, which can significantly increase muscle functions in older adults when planned and followed up properly. Future studies should aim for larger sample sizes. This will help to clarify the distinct contributions of diet and exercise to health outcomes in older adults.

The ASMI is used to measure muscle mass in the evaluation of sarcopenia. Evidence shows that muscle strength and muscle mass do not always correlate. According to EWGSOP2, muscle strength measurement is the first step and provides the most reliable criterion for analyzing muscle function; low muscle strength together with low muscle mass indicates confirmed sarcopenia [4]. We found that 90% of the patients in both groups had low HGS, and 20% of the patients in the PD group had low ASMIs in the initial analysis. All of our patients reached normal values after the study interventions. Our data show a significant improvement in muscle strength and physical performance in both groups at the end of the study. A systematic review and meta-analysis in older adults showed that malnourished patients had significantly lower HGS [28]. In a prospective observational study by Abizanda et al., oral nutritional therapy combined with a physical exercise regimen in 69 frail older adults did not lead to a significant improvement in muscle strength after 12 weeks [29]. However, in a meta-analysis with 24 randomized and non-randomized studies, a significant improvement in HGS was observed with physical exercise [30]. Another randomized controlled trial involving 811 participants aged ≥65 years with MN risk showed that patients receiving oral nutritional supplementation had a significant improvement in HGS at day 180, compared to placebo [31].

In our study, muscle function tests showed improvements such as increased muscle strength and decreased TUG times in both groups at the end of the study. We believe that there could be some reasons for the lack of additional effects of exercise on muscle mass in our study. First, this could be related to the low number of participants, as a limitation of the study. Second, although the patients were followed up through phone calls, their exercise adherence in the home could have been low. Third, low muscle mass is related to many factors in older adults, such as cellular senescence, immobility, malnutrition, neurological denervation, atrophy of the muscle fibers, decreased IGF-1, inflammaging, atherosclerosis, chronic diseases, and drugs. Stay-at-home restrictions during the COVID 19 pandemic were related to significant muscle loss in the elderly. Thus, a 12-week exercise plan at home might not be enough to improve muscle mass.

Strasser et al. followed 20 older adults with MN and MN risk during an orthopedic rehabilitation program, and they compared the effects of protein-enriched foods and drinks with those of standard care. Despite a significant increase in protein intake in the intervention group, they did not find any significant improvement in HGS, body composition, or physical function [32]. In our study, each patient was provided an individualized diet considering the daily needs of the patient, their own nutritional habits, socioeconomic status, and the education of the caregivers. Moreover, we also measured QoL changes after the nutrition and physical exercise interventions. According to our results, a well-planned personalized diet with or without physical exercise could increase the muscle strength, physical performance, and QoL in our patients. Moreover, the EQ5D VAS score and ASM measures of the whole study group were found to be correlated at the end of the study.

In our study, the TUG times of the patients were significantly improved in both groups. The recovery improvement rates were similar in the PD and PDE groups. Steffen et al. showed that the TUG test was highly reliable for measuring physical performance in older adults [33]. Ramsey et al. examined the relationship between MN and physical performance in 286 older adults and showed that the TUG test was significantly associated with MN [34].

Previous reports indicated a possible relationship between nutritional status and QoL in older adults. A cross-sectional analytical observational study that included 802 patients showed a significantly negative association between BMI and quality of life [35]. A cohort study with 1326 adults (median age: 72) analyzed the relationship between BMI and QoL, and found a significant difference in QoL scores between different BMI groups, with the highest scores at normal BMIs and the lowest in obesity [36]. In a prospective observational study, oral nutritional support was associated with an increase in EQ-5D-VAS scores [29].

Kayn. Lu et al. found that the TUG times of patients were negatively correlated with EQ-5D scores, indicating that higher TUG times (meaning lower physical performance) were associated with lower QoL in both sexes [37]. According to our results, MN treatment with a personalized diet increased QoL. The average EQ-5D scores were significantly increased in the diet PD group, and the average VAS scores were significantly increased in both groups. A negative correlation was found between TUG times and EQ5DVAS score changes after 12 weeks of intervention in the whole study group.

Our study had strengths and limitations. Previous work predominantly examined using oral nutritional supplements to treat malnutrition in older adults, while our work focused on the efficacy of a personalized diet, for which previous evidence was limited. The strengths of our study included its prospective, randomized, and controlled design, with objective and reliable measurements of nutritional status, muscle mass and function, and quality of life.

## 5. Conclusions

A personalized diet with or without physical exercise therapy was associated with improved nutritional status, physical performance, and QoL. However, this study also had some limitations, such as its small sample size. The patients in our study were over the age of 65, and due to the COVID-19 pandemic, some restrictions were imposed on this age group, such as a curfew during certain hours and a ban on using public transportation. In addition, because our study excluded those with acute illnesses, few individuals could apply to outpatient clinics at that time, and it was difficult to find volunteers for the study.

Larger old-aged cohorts are required to better understand the effects of a personalized diet combined with structured physical exercise on malnutrition treatment, physical performance, and QoL.

## Figures and Tables

**Table 1 nutrients-17-00675-t001:** Characteristics of the patients.

	PD Group (*n* = 10)	PDE Group (*n* = 10)	Total (*n* = 20)	*p*–Value
Gender (*n*)				
Male	6	5	11	0.65
Female	4	5	9	
Age (years)				
Male	72.7 ± 8.0	72.6 ± 4.7	72.6 ± 6.4	0.97
Female	71.0 ± 4.5	72.4 ± 3.8	70.7 ± 3.9	
Chronic diseases				
Hypertension	3	2	5	0.26
Diabetes	6	7	13	
Heart failure	1	2	3	
CKD	1	0	1	
Coronary artery disease	1	1	1	
# of CD (median, min-max)	Median	Median		
Nutrition status				
MN risk *	4 (%40)	4 (%40)		
MN **	6 (%60)	6 (%60)		
Initial mean BMI (kg/m^2^)	25.9	29.2		0.247
Initial mean ASMI (kg/m^2^)	7.7	8.5		0.280

CKD: chronic kidney disease; PD: personalized diet; PDE: personalized diet with planned exercise; * MNA-SF score 8–11; ** MNA-SF score ≤ 7; BMI: body mass index; ASMI: appendicular skeletal muscle mass; #: number.

**Table 2 nutrients-17-00675-t002:** BMI, MNA, handgrip strength, ASMI, TUG test, EQ-5D, and EQ–5D VAS measurements of the patients during the study.

	PD Group							PDE Group						
**BMI (kg/m^2^)**	**M**	**SD**	**Mdn**	**IQR**	**Z**	** *p* **	**±%**	**M**	**SD**	**Mdn**	**IQR**	**Z**	** *p* **	**±%**
Initial	25.9	5.0	26.7	7.3	−2.80	0.005	0	29.2	4.7	28.7	8.0	−2.45	0.01	0
4. wk	26.9	4.7	27.5	7.3			3.9	29.4	4.4	29.2	6.8			0.7
8. wk	27.0	4.5	27.7	7.0			4.2	29.4	4.1	29.3	6.7			0.7
12. wk	27.6	4.6	28.9	7.4			6.6	30.1	4.9	29.4	7.0			3.1
**MNA-SF** **Test (score)**	**M**	**SD**	**Mdn**	**IQR**	**Z**	** *p* **	**±%**	**M**	**SD**	**Mdn**	**IQR**	**Z**	** *p* **	**±%**
Initial	7.1	1.7	7.0	3	−2.84	0.005	0	7.7	1.1	7.0	1	−2.83	0.005	0
4. wk	10.1	1.7	10.5	3			42.3	10.4	1.1	11.0	2			35.1
8. wk	11.7	0.5	12.0	1			64.8	11.7	0.8	12.0	1			51.9
12. wk	13.0	0.7	13.0	1			83.1	13.5	0.9	13.5	1			75.3
**EQ-5D index (score)**	**M**	**SD**	**Mdn**	**IQR**	**Z**	** *p* **	**±%**	**M**	**SD**	**Mdn**	**IQR**	**Z**	** *p* **	**±%**
Initial	0.75	6.3	0.76	9	−2.57	0.01	0	0.70	15.5	0.75	14	−0.14	0.89	0
4. wk	0.73	4.0	0.73	3			2.4	0.71	11.6	0.73	7			1.7
8. wk	0.80	8.1	0.80	9			6.7	0.72	11.9	0.73	8			2.6
12. wk	0.82	8.0	0.80	8			9.4	0.74	13.1	0.75	14			5.3
**EQ-5D** **VAS (score)**	**M**	**SD**	**Mdn**	**IQR**	**Z**	** *p* **	**±%**	**M**	**SD**	**Mdn**	**IQR**	**Z**	** *p* **	**±%**
Initial	67.0	14.2	70.0	30	−2.36	0.02	0	65	13.5	65	23	−2.39	0.02	0
4. wk	73.5	6.7	72.5	10			9.7	71	11.0	70	20			9.2
8. wk	73.0	6.7	70.0	10			9	73	9.5	70	13			12.3
12. wk	79.0	5.7	80.0	3			17.9	78	6.3	80	10			20.0
**ASMI (kg/m^2^)**	**M**	**SD**	**Mdn**	**IQR**	**Z**	** *p* **	**±%**	**M**	**SD**	**Mdn**	**IQR**	**Z**	** *p* **	**±%**
0	7.7	1.5	7.6	2.5	−1.68	0.09	0	8.5	1.0	8.4	1.4	−0.36	0.72	0
4. wk	8.5	0.8	8.4	1.1			10.4	8.7	1.3	8.6	1.5			2.4
8. wk	8.3	0.8	8.4	1.5			7.8	8.6	1.1	8.5	1.1			1.2
12. wk	8.4	0.9	8.5	1.4			9.1	8.6	1.2	8.3	1.2			0
**HS (kg)**	**M**	**SD**	**Mdn**	**IQR**	**Z**	** *p* **	**±%**	**M**	**SD**	**Mdn**	**IQR**	**Z**	** *p* **	**±%**
0	20.9	5.3	19.0	11	−2.81	0.005	0	20.3	8.5	19.0	14.0	−2.81	0.005	0
4. wk	23.5	6.4	22.0	11			12.4	23.4	8.9	23.0	13.0			15.3
8. wk	25.4	5.1	26.5	9			21.5	26.2	7.7	25.0	10.0			29.1
12. wk	27.5	6.1	29.0	11			31.6	27.3	8.3	25.5	13.0			34.5
**TUG Test (s)**	**M**	**SD**	**Mdn**	**IQR**	**Z**	** *p* **	**±%**	**M**	**SD**	**Mdn**	**IQR**	**Z**	** *p* **	**±%**
0	13.6	2.9	12.9	3.2	−2.81	0.01	0	15.0	4.9	14.3	4.3	−2.81	0.01	0
4. wk	11.9	2.1	11.9	3.3			−12.5	13.4	4.0	12.2	3.4			10.7
8. wk	11.3	1.8	10.6	2.7			−16.9	12.5	3.0	11.5	3.1			16.7
12. wk	10.8	1.7	10.0	2.6			−20.6	11.9	2.7	11	3.2			20.7

BMI: body mass index; MNA-SF: Mini Nutritional Assessment—Short Form; EQ-5D: EuroQol-5 Dimensions; ASMI: appendicular skeletal muscle mass index; HS: handgrip strength; TUG: Timed Up and Go test; M: mean; Mdn: median; SD: standart deviation; IQR: interquartile range; Z: z-score or test statistic; wk: weeks.

**Table 3 nutrients-17-00675-t003:** Correlation between the QoL score and other variables’ changes after 12 weeks of intervention in the whole study group (Spearman correlations).

	ΔBMI	ΔMNA-SF	ΔHGS	ΔASM	ΔASMI	ΔTUG
ΔEQ-5D VAS (12th w—initial)	−0.034	0.251	−0.190	0.184	0.216	−0.550
*p* value	0.886	0.285	0.423	0.438	0.361	0.012 *
ΔEQ-5D index (12th w—initial)	0.258	0.177	0.066	−0.67	−0.154	0.227
*p* value	0.273	0.455	0.782	0.780	0.517	0.335

ASM: appendicular skeletal muscle mass; ASMI: appendicular skeletal muscle mass index; BMI: body mass index; HGS: handgrip strength; MNA-SF: Mini Nutritional Assessment—Short Form; TUG: Timed Up and Go test; w: week; * *p* < 0.05.

## Data Availability

The data in our study are kept confidential due to ethical reasons related to the patients. The personalized diets created individually for each patient make sharing difficult due to the data’s complexity. Study was retrospectively registered, with clinical trial number NCT06823739 on 12 February 2025.

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
