# Peer review of "Personalized Diet With or Without Physical Exercise Improves Nutritional Status, Muscle Strength, Physical Performance, and Quality of Life in Malnourished Older Adults: A Prospective Randomized Controlled Study"

_nutrients, 2025, doi:10.3390/nu17040675_

Round 1
Reviewer 1 Report
Comments and Suggestions for Authors
This study addresses an important topic and employs a relevant intervention design. However, the small sample size, baseline imbalances, and limited generalizability raise concerns about the validity and reliability of the findings.
Methods. While randomization was applied based on admission dates, small sample sizes can lead to imbalances between groups, which might affect the validity of the comparisons. A clearer justification for the small sample size, ideally supported by a power analysis, and framing the study as preliminary or exploratory should be provided.
Point 2.3. Please explain how patients were weighed and measured. How was the BIA analysis done? Were these measurements made the same way every time?
Points 2.5. and 2.6. Sentences including "the same two dietitians" or “the same two staff physioterapists” should be rephrased to clarify the reference, such as "two appointed dietitians" or “two appointed physiotherapists”. Additionally in point 2.5., simplifying "which was specifically designed based on the patients’ needs" to "tailored to the patients' needs" would enhance clarity.
Point 2.6. Describe and cite Borg scale.
Statistical analysis. Please maintain consistency in terminology by replacing ambiguous phrases like "number of the universe" with precise terms such as "sample size" or "population size". Additionally, the incorrect statement "the standard error is standard deviation of the universe" should be corrected, as standard error is derived by dividing the standard deviation by the square root of the sample size. Redundancy in explanations, such as the repeated description of the Mann-Whitney U Test, should be eliminated to streamline the text. Furthermore, the rationale for using Fisher's exact test for a population of 20 should be briefly clarified to avoid potential misinterpretations and ensure statistical justifications are transparent.
Table 1. Some results are missing in the fifth line.
Table 2. The PDE group starts with a higher BMI and ASMI than the PD group, which could confound the interpretation of results. While weight gain (reflected by BMI) may be a desirable outcome for malnourished elderly individuals, the difference in trends between the two groups raises questions about whether exercise provides additional benefits beyond weight stabilization. Could you discuss this issue and make relevant conclusions?
Discussion. A detailed discussion of baseline imbalances and their potential impact on the results should be provided.
Limitations must be revised more profoundly.
Comments on the Quality of English Language
The language is comprehensible but requires some improvement.
Author Response
Comments 1: This study addresses an important topic and employs a relevant intervention design. However, the small sample size, baseline imbalances, and limited generalizability raise concerns about the validity and reliability of the findings.
Methods. While randomization was applied based on admission dates, small sample sizes can lead to imbalances between groups, which might affect the validity of the comparisons. A clearer justification for the small sample size, ideally supported by a power analysis, and framing the study as preliminary or exploratory should be provided.
Response 1: The primary aim of this study is to explore the feasibility and potential effects of personalized diet interventions, with or without exercise, in malnourished elderly individuals. As a preliminary study, it provides initial insights and identifies trends that warrant further investigation. Recruiting participants from a specific population, such as older adults, can be challenging due to strict inclusion criteria and the health status of the volunteers. Randomizing patients based on their date of admission is a practical approach that simplifies the logistics of enrollment and randomization, especially in a clinical setting where immediate assignment to an intervention is necessary. While not the most robust form of randomization, using admission dates helps to avoid selection bias by ensuring that all eligible patients are considered for inclusion as they present, rather than being selectively chosen. Conducting a power analysis requires estimates of effect sizes, which might not have been available or reliable for this particular population or intervention type.
When evaluating the statistical power of the study, a priori power analysis was conducted using G*Power (version 3.1). The analysis was based on an ANOVA: Repeated measures, within-between interaction design to determine the required sample size for adequate power. In the context of our findings variable, the observed power for the within-subject factor (time) was 0.81, with a partial eta squared (η²) of 0.07, indicating a medium effect size. For the interaction effect (time × group), the partial eta squared (η²) of 0.072, also reflecting a medium effect size. The input parameters included an effect size f=0.278, alpha error probability (α) = 0.05, desired power (1−β) = 0.8, two groups (intervention and control), four repeated measurements, and a correlation among repeated measures of 0.5. The nonsphericity correction factor (ϵ) was set to 1. The analysis determined a total required sample size of 20 participants (10 per group), with an actual power of approximately 0.823. These findings suggest that a minimum of 10 participants per group is sufficient to conduct an adequately powered trial (1−β=0.8, α=0.05) in this rehabilitation setting.
Changes applied at page 2, point 2.1, line 76-96
Comments 2: Point 2.3. Please explain how patients were weighed and measured. How was the BIA analysis done? Were these measurements made the same way every time?
Response 2: Patients were weighed with BIA (Tanita MC 780 MA, Japan) at each visit. Measurements were done with the same device in the Clinical Nutrition and Microbiota Research Laboratory, Istanbul Faculty of Medicine. The device was always in the same place during all visits. During the measurement, it was checked that the patients took off their shoes and socks, did not wear heavy or tight clothes, did not have moisturizers such as creams on their hands and feet, and did not carry transmitting devices such as mobile phones.
Changes applied at page 3, point 2.3, line 108-114
Comments 3: Points 2.5. and 2.6. Sentences including "the same two dietitians" or “the same two staff physioterapists” should be rephrased to clarify the reference, such as "two appointed dietitians" or “two appointed physiotherapists”. Additionally in point 2.5., simplifying "which was specifically designed based on the patients’ needs" to "tailored to the patients' needs" would enhance clarity.
Response 3: corrections applied at page number 3, point 2.5, line 131-132 and page number 4, point 2.6, line 169-170
Comments 4: Point 2.6. Describe and cite Borg scale.
Response 4: The Borg Scale is a tool used for measure an individual’s perceived level of effort during physical activity. It is based on how hard a person feels their body is working. The scale ranges from 6 to 20, with each number corresponding to a level of intensity. In this scale 6 means no exertion at all and 20 means maximal exertion.
Describing and cite added at page 4, point 2.6, line 173-177
Comments 5: Statistical analysis. Please maintain consistency in terminology by replacing ambiguous phrases like "number of the universe" with precise terms such as "sample size" or "population size". Additionally, the incorrect statement "the standard error is standard deviation of the universe" should be corrected, as standard error is derived by dividing the standard deviation by the square root of the sample size. Redundancy in explanations, such as the repeated description of the Mann-Whitney U Test, should be eliminated to streamline the text. Furthermore, the rationale for using Fisher's exact test for a population of 20 should be briefly clarified to avoid potential misinterpretations and ensure statistical justifications are transparent.
Response 5: Quantitative variables were expressed as mean, median, and interquartile range (IQR) for continuous data, and as percentage (%) and frequency (n) for categorical data. Since the data did not follow a normal distribution, non-parametric tests were used for comparisons.
The Kruskal-Wallis test was applied to compare more than two non-parametric groups. For pairwise comparisons, the Mann-Whitney U test was used as a post-hoc test. Spearman’s correlation test measured the correlation between variables, while the chi-square test was used to compare categorical variables. Fisher's Exact Test was chosen because it provides accurate results for small sample sizes (n<30) and is particularly suitable when expected frequencies in contingency tables are less than 5.
All results were evaluated within a 95% confidence interval, and p<0.05 was considered statistically significant. Statistical analyses were performed using IBM SPSS-21 (Statistical Package for Social Sciences version 21.0).
Changes applied at page 5, point 2.8, line 203-215
Comments 6: Table 1. Some results are missing in the fifth line.
Response 6: Fixed the missing results at Table 1.
Comments 7: Table 2. The PDE group starts with a higher BMI and ASMI than the PD group, which could confound the interpretation of results. While weight gain (reflected by BMI) may be a desirable outcome for malnourished elderly individuals, the difference in trends between the two groups raises questions about whether exercise provides additional benefits beyond weight stabilization. Could you discuss this issue and make relevant conclusions?
Response 7: Higher baseline BMI and ASMI values ​​are seen in the PDE group. But at the beginning of the study, the PD and PDE groups were not statistically different from each other in terms of mean BMI and ASMI. The p values of this analysis were added to Table 1. However, due to their higher starting muscle mass, these improvements might not translate into significant BMI changes compared to the PD group.
Comments 8: Discussion. A detailed discussion of baseline imbalances and their potential impact on the results should be provided.
Response 8: In our study, the initial mean BMI and ASMI of the PDE group were higher than the PD group, which were statistically non-significant (Table 1). Despite the small number of patients, the study provided important insights into the role of diet and exercise interventions in improving health outcomes among older adults. Probably with a higher number of participants, PDE could make difference in terms of muscle mass. However, the results of this preliminary study highlighted the importance of the personalized diet, which can add significant increase to the muscle functions in the older adults, when planned and followed-up properly. Future studies should aim for larger sample sizes. This will help clarify the distinct contributions of diet and exercise to health outcomes in older adults.
Changes applied at page 9, line 313-322
Comments 9: Limitations must be revised more profoundly.
Response 9: The patients in our study were over the age of 65, and due to the Covid-19 pandemic, some restrictions were imposed on this age group, such as a curfew at certain hours and a ban on using public transportation. In addition, since our study excluded those with acute illnesses, few individuals could applied to outpatient clinics at that time, and it was difficult to find volunteers for the study.
Changes applied at page 10, point 5., line 391-396
Reviewer 2 Report
Comments and Suggestions for Authors
Thank you for the interesting manuscript describing the relevance of personalized diets (instead of ONS). Attention for personalized seems logical, but the effects of improved daily diets for frail and malnourished older adults are not studied well enough to push for strong recommendations and guidelines. My comments for this manuscript are:
- Describe the diet in more detail. What kind of food products, animal and plant-based foods and protein, en% protein, distribution of energy and protein over the meals etc. The personalized diets are unique for this study.
- Describe the strength exercises in methods. In results section, the results of the statistical comparison between PD and PDE groups should be mentioned. Explain in discussion in more detail why the exercise did not show an additional effect.
- Describe the intervention effect on body weight instead of (at least in addition to) BMI. The effects are on body weight, not on height. Other studies show a BMI of approx. of 26/27 as mean instead of 'normal' BMI 20-25. Use other studies to put the BMI in perspective.
- Mention the changes in fat free mass and fat mass detected by BIA during the intervention in addition to ASMI. These should be associated to changes in body weight (see above) and the effects on body composition should be discussed.
- In methods, it is stated that TUG > 20 sec is associated with poor performance. The TUG values are 13,6 and 15,0 sec, but I do not understand how 100% and 90% low physical performance was found in the groups.
Author Response
Comments 1: Thank you for the interesting manuscript describing the relevance of personalized diets (instead of ONS). Attention for personalized seems logical, but the effects of improved daily diets for frail and malnourished older adults are not studied well enough to push for strong recommendations and guidelines. My comments for this manuscript are:
Describe the diet in more detail. What kind of food products, animal and plant-based foods and protein, en% protein, distribution of energy and protein over the meals etc. The personalized diets are unique for this study.
Response 1: Personalized diet was prepared by the two appointed dietitians of the Clinical Nutrition Team of the Hospital, tailored to the patients' needs (according to daily energy and protein requirements, which were calculated with 25-30 kcal/kg and 1.0-1.2 g/kg, respectively) [16], lifestyle, nutritional preferences and medical diagnoses. Physiological stress and the physical activity level of the patients were also taken into consideration.
45-55% of total calories came from carbohydrates, 20-35% from fat and diet included 20-30 gr fiber/daily [17].
While preparing the diet plan, the patients' socioeconomic status, palate, social and psychological factors were taken into consideration. Protein sources were diversified, with animal-based proteins (chicken, fish, red meat, turkey) and plant-based proteins (legumes). They were asked which protein source they usually consume and which one they prefer more in terms of taste. For example, it was explained to patients who had difficulty consuming meat could prefer legumes, eggs or dairy products instead of meat. Patients were told to consume fish at least twice a week.
Each breakfast featured eggs and fermented dairy products. The fat content predominantly comprised mono- and polyunsaturated fatty acids, from olive oil, sunflower oil, fish, and nuts. Carbohydrates were selected to have a low glycemic index, favoring whole grains while excluding white flour entirely.
While preparing the diet plan, the people whom the patient lived with, were also consulted. It was questioned whether the patient himself/herself or the people he/she lived with mainly cooked the meals at home. Thus, main meals were discussed with the primer person cooking at home.
The inclusion of fruits and vegetables aimed to maximize antioxidant diversity and color variety. To enhance calorie intake and dietary adherence, milk-based and fruit-based desserts were incorporated twice weekly. Hydration goals were set at a minimum of 30 ml/kg/day with seasonal variations. For patients struggling to meet these goals, additional fluids were provided through freshly squeezed juices and unsweetened homemade compotes. Tea and coffee consumption were limited to one cup daily. For individuals requiring caffeine restrictions due to conditions such as hypertension, cardiac disease, arrhythmia, or anxiety, decaffeinated coffee was recommended.
Daily salt intake was restricted to 5–6 grams, and for patients with previously high salt consumption, herbs and spices were used to enhance flavor and improve adherence. To support meal planning and encourage dietary compliance, participants received education on the "healthy plate model," promoting variety and balanced nutrition[16]. Diet compliance was asked with the regular phone calls in every week and during the visits.
Changes applied at page 3, point 2.5., line 138-139 and page 4, line 140-164
Comments 2: Describe the strength exercises in methods. In results section, the results of the statistical comparison between PD and PDE groups should be mentioned. Explain in discussion in more detail why the exercise did not show an additional effect.
Response 2: The exercise program in PDE group was designed to be performed at home without requiring any specialized equipment. It aims to improve physical health, balance, and coordination while maintaining muscle strength and flexibility. Patients performed brist walking at least three times a week for a duration of 30 minutes per session. It also included balance practice and coordination exercises for another 10 minutes. These include: Standing on tiptoes, balancing on one foot while holding onto a chair for support, walking in a straight line as if on a narrow beam. Finally, the exercise session ended with a 10-minute cool-down, focusing on relaxing muscles and performing gentle stretches to improve flexibility and reduce tension. All of the exercises were both explained and given in a figured charts, to the patients or the relatives.
Changes applied at page 4, point 2.6., line 178-189
In our study, muscle function tests were improved in both groups, like increased muscle strength and decreased TUG times in both groups at the end of the study. We believe that there could be some reasons for the lack of additional effects of exercise on muscle mass in our study. First, this could be related with the low number of the participants, as a limitation of the study. Second, although the patients were followed up through phone calls, their exercise adherence in the home could be low. Third, low muscle mass is related with many factors in the older adults, such as cellular senescence, immobility, malnutrition, neurological denervation, atrophy of the muscle fibers, decreased IGF-1, inflammaging, atherosclerosis, chronic diseases, drugs, etc. Stay at home restrictions during COVID 19 pandemics related with significant muscle loss in the elderly. So 12 weeks of exercise plan at home might be not enough to improve mucsle mass.
Changes applied at page 9, line 340 -350
Comments 3: Describe the intervention effect on body weight instead of (at least in addition to) BMI. The effects are on body weight, not on height. Other studies show a BMI of approx. of 26/27 as mean instead of 'normal' BMI 20-25. Use other studies to put the BMI in perspective.
Response 3: During the study, the mean weight change of the patients in the PD group was statistically significant (p=0.005). Although it was increased at the end of the study in PDE group, that difference could not reach statistical significance (p=0.074). This may be due to limited number of patients in the study and increased energy expenditure with the exercise program in PDE group.
Although various BMI ranges have been proposed over the years for what is considered normal in the geriatric population, the prevailing consensus suggests that the normal BMI in older adults differs from that observed in younger populations. A recent meta-analysis has indicated that the BMI range of 23-28 kg/m² is the most strongly associated with lower mortality and disability, and as such, may be considered optimal for the geriatric population.
Changes applied and reference added at page 9, line 306-308
Comments 4: Mention the changes in fat free mass and fat mass detected by BIA during the intervention in addition to ASMI. These should be associated to changes in body weight (see above) and the effects on body composition should be discussed.
Response 4: Unfortunately, parameters of fat-free mass and fat mass could not be obtained retrospectively. BIA measurements on fat mass were not included in the analysis.
Comments 5: In methods, it is stated that TUG > 20 sec is associated with poor performance. The TUG values are 13,6 and 15,0 sec, but I do not understand how 100% and 90% low physical performance was found in the groups.
Response 5: Due to the definition in the methods section, the patients' initial mean TUG values cannot be classified as low physical performance. This incorrect statement in the results section is removed.
Round 2
Reviewer 2 Report
Comments and Suggestions for Authors
Thanks for the comments. I agree with the comments. It is a pity that fat mass data are not available anymore. I would prefer to see body weight data as well. I agree with the comments regarding BMI.